# Assessing Habitat Suitability for *Phloeosinus aubei* Perris in China: A MaxEnt-Based Predictive Analysis

**DOI:** 10.3390/insects16060576

**Published:** 2025-05-29

**Authors:** Sabbir Ahmad, Danping Xu, Xinqi Deng, Zhipeng He, Habib Ali, Zhihang Zhuo

**Affiliations:** 1College of Life Science, China West Normal University, Nanchong 637002, China; ahmad.sabbir76@gmail.com (S.A.); xudanping@cwnu.edu.cn (D.X.); deng.xinqi@foxmail.com (X.D.); zhipeng_hh@foxmail.com (Z.H.); 2Department of Agriculture Engineering, Khwaja Fareed University of Engineering and Information Technology, Rahim Yar Khan 64200, Pakistan; habib_ali1417@yahoo.com

**Keywords:** *Phloeosinus aubei*, invasive species management, species distribution modeling, suitable habitat, ecological impacts

## Abstract

Climate change is shifting where species can survive, and this study explored how it may affect the spread of *Phloeosinus aubei*, a tree-damaging beetle, in China. Using a computer model, we predicted where this beetle could live now and in the future by analyzing weather patterns and land features. We found that temperature and rainfall are the main factors influencing its spread. By the 2050s, areas highly suitable for the beetle could grow by over 80%, especially in warmer, wetter parts of southwestern China. This poses a threat to forests, as the beetle harms already weakened trees and can cause serious damage to ecosystems. This study highlights the need for early detection and smarter forest management to protect vulnerable areas. These findings can guide future planning to reduce the risks of invasive pests under climate change.

## 1. Introduction

Understanding the global spread of *Phloeosinus aubei* (Coleoptera: Curculionidae: Scolytinae), commonly known as the cedar bark beetle, is essential for assessing both its ecological and economic impacts [1]. Native to the Mediterranean region and central Europe, *P. aubei* has increasingly expanded its geographical range, a development largely driven by climatic variability, changes in land use, and the availability of suitable host populations [2]. While climatic factors are critical, anthropogenic activities such as timber trade and urbanization likely accelerate *P. aubei’s* spread, particularly in fragmented landscapes. This bark beetle, measuring between 2 and 2.7 mm in length, is readily identified by its punctate dorsum, small setae (hairs), black thorax, and brownish elytra (wing covers) with longitudinal stripes [3]. Although considered a secondary invader, *P. aubei* is a secondary bark beetle pest, meaning it predominantly infests trees already stressed by factors such as drought, disease, or prior insect damage. Unlike primary pests, which attack healthy trees, *P. aubei* accelerates the decline of weakened hosts, leading to cascading ecological impacts, such as reduced tree growth, increased mortality, and altered forest structure [4]. The introduction and spread of *P. aubei* in novel habitats present substantial challenges to forest health management and biodiversity conservation [5,6]. As an opportunistic pest, it exploits weakened or stressed trees, which, in turn, results in reduced tree growth rates, increased mortality, and eventually the degradation of forest stands [7]. In ecosystems where *P. aubei* establishes, it may alter the species composition of forests, potentially displacing native insect populations and affecting plant–pollinator interactions [8]. This can lead to broader ecological consequences, including altered nutrient cycling, changes in forest structure, and diminished wildlife habitat quality.

MaxEnt has become a widely adopted tool in ecological and biogeographic research because of its flexibility and robustness [9,10]. It can integrate both continuous (e.g., temperature and precipitation) and categorical (e.g., land cover types) environmental variables to predict species distributions under current or future climate scenarios [11]. Moreover, MaxEnt excels in handling situations where occurrence data are sparse or incomplete, a common challenge when studying new or invasive species. Its ability to extrapolate from a small number of occurrence records by leveraging a wide range of environmental predictors makes it an ideal tool for forecasting species spread, especially for species like *P. aubei*, which may be in the early stages of invasion in regions such as China [12].

Given the growing concern over the impacts of *P. aubei*, early detection and effective management are critical to preventing widespread damage [13]. Ecological modeling approaches have become a cornerstone for assessing the potential distribution of the species in novel habitats. These models can predict areas that are climatically suitable for the beetle’s survival and expansion, providing valuable information for developing monitoring and management strategies. In particular, Species Distribution Models (SDMs) allow researchers and forest managers to estimate potential habitat suitability based on environmental and climatic data, making them vital tools for proactive forest health management [14].

To better understand the current distribution of *P. aubei* and its potential spread across China, it is essential to visualize the geographic occurrence points [15]. In China, *P. aubei* is currently documented across latitudes 20° N–50° N and longitudes 80° E–140° E (Figure 1), with higher prevalence in eastern provinces. These regions align with temperate and subtropical zones, where the beetle’s primary host species, cypress (*Cupressus* spp.), thrives. This established distribution highlights areas already at risk and serves as a baseline for assessing future habitat shifts under climate change.

This study aims to expand the understanding of *P. aubei*’s potential distribution within China by applying the MaxEnt model to estimate its habitat suitability under future climate scenarios. By integrating various environmental and climatic factors, we seek to identify regions at high risk of infestation. These regions are not only essential for guiding management strategies but also for prioritizing areas for monitoring, early detection, and intervention. The results of this study could inform the development of adaptive forest management strategies that mitigate the spread of *P. aubei* and protect vulnerable ecosystems. Moreover, the insights gained from this research could be applied to other countries facing similar ecological challenges, offering a more global perspective on managing invasive pests in the face of climate change.

## 2. Materials and Methods

### 2.1. Species Occurrence Data and Bioclimatic Variables 

In this study, we downloaded 19 historical global bioclimatic variables from the Global WorldClim database (http://www.worldclim.org/download, accessed on 10 June 2024). The terrain variables are available from the National Oceanic and Atmospheric Administration (NOAA, https://www.noaa.gov/, accessed on 25 January 2022). The environmental variable data were snipped and extracted using the Mask Extract tool in ArcGIS 10.4 to retain the needed data range and converted to the ASCII format. Furthermore, a provincial-rank directorial chart at a scale of 11.4 million was downloaded from the National Fundamental Geographic Information System of China (https://www.ngcc.cn/) as the analysis base map (GS2024 0650). These bioclimatic variables affect the growth and development of *P. aubei*. The future climate data for the 2050s and 2090s were obtained by querying the Climate Change, Agriculture, and Food Security website (CCAFS, https://ccafs.cgiar.org/, accessed on 29 January 2024).

Among the four greenhouse gas concentration pathways reported by the IPCC, three scenarios—SSP1-2.6 (low emissions), SSP2-4.5 (medium emissions), and SSP5-8.5 (high emissions)—were selected based on their wide usage in climate impact studies and their ability to represent a broad spectrum of possible future socioeconomic and emission trajectories. These scenarios provide a comprehensive basis for simulating and predicting the future distribution of *P. aubei* under varying climate conditions. Pearson correlation analysis was conducted in SPSS 26.0 to assess the direct correlation among climatic variables. Climatic variables with correlation coefficients higher than 0.8 were removed to reduce the influence of overfitting on the model, ensuring the robustness and accuracy of the model. Finally, the distribution forecasting model was constructed using eight selected bioclimatic variables—alt, bio1, bio12, bio13, bio15, bio3, bio6, and bio8—as shown in Table 1.

### 2.2. Model Settings and Operation

To limit the influence of spatial autocorrelation and selection bias on the final casting, the SDM Toolbox was used to reuse the spare incident data, ensuring that only one distribution point was involved in every 4.5 km × 4.5 km grid cell [16]. This sparse point approach was linked with the diagnosis of the environmental data to decrease the composition of distribution points caused by spatial autocorrelation and avoid model overfitting [17]. The related and filtered data were imported into MaxEnt 3.4.4 software to prepare for the evaluation and model creation of bioclimatic variables [17]. Leap experiments were undertaken to estimate the bioclimatic factors. The obtained model aimlessly selected 25% of the species incidence data as the test set, with the remaining 75% utilized as the training set. The model was estimated and verified using 10 reiterations and up to 5000 duplications [18]. This strategy is intended to reduce any biases coming from inadequate background data. Furthermore, the essential task of finding the most suitable failure parameter was rigorously handled utilizing the Bootstrap technique [19]. Model accuracy was evaluated using the Receiver Operating Characteristic (ROC) curve and the Area Under the Curve (AUC) metric, where an AUC < 0.7 indicates low predictive accuracy, 0.7 ≤ AUC < 0.8 suggests moderate reliability, 0.8 ≤ AUC < 0.9 reflects good accuracy, and AUC ≥ 0.9 denotes high predictive performance.

### 2.3. Optimization of Model Parameters

In this study, the MaxEnt model was applied to estimate the distribution of *P. aubei* for the present and future projections in the 2050s and 2090s.

To determine the optimal model parameters, the regularization multiplier (RM) value was tested across a range from 0.1 to 4, with a step size of 0.1. The R package ENMeval, developed by Muscarella et al. [20], was used to fine-tune the model parameters. This package evaluates various combinations of feature classes and regularization multipliers to identify the most suitable model setting. Specifically, in the R package “ENMeval”, regularization multiplier settings were paired with different point class combinations, generating 1240 candidate models for evaluation.

The selection of optimal parameter combinations was based on statistical significance, omission rates below 5%, and a delta AICc (Akaike Information Criterion corrected) value of ≤2. This criterion ensures that the chosen parameters enhance the model’s predictive accuracy while maintaining a low omission rate, thereby improving model resilience. The delta AICc ≤ 2 threshold helps identify models with near-optimal performance, preventing unnecessary model complexity. Overall, the results highlight the effectiveness of the optimization procedures employed and confirm the strong predictive capability of the final MaxEnt model (ENMeval). This robust statistical approach provides reliable predictions while minimizing the risk of overfitting. The final optimized model, labeled “kuenm”, demonstrated superior performance.

### 2.4. Suitable Area Division and Model Accuracy Evaluation

Based on the ROC curve generated by the Maxent model and the calculated AUC value, the AUC value obtained from the averaged model is 0.901 (Figure 2).

The True Skill Statistic (TSS) was also used, with values above 0.7 indicating strong predictive accuracy and values below 0.5 suggesting poor performance. MaxEnt output ASC files were converted to raster format in ArcGIS, and habitat suitability was categorized based on the IPCC probability classification approach: highly suitable (*p* ≥ 0.66, red), moderately suitable (0.33 ≤ *p* < 0.66, orange), low suitability (0.05 ≤ *p* < 0.33, yellow), and unsuitable (*p* < 0.05, white).

## 3. Results

### 3.1. Key Climatic Drivers of P. aubei Habitat Suitability

To understand the key environmental factors influencing the habitat suitability of *P. aubei*, it is essential to assess the contribution of various bioclimatic variables to the MaxEnt model [21]. Table 1 summarizes the results of the model’s analysis, detailing the relative importance of each variable in predicting the species’ distribution. By examining both the percent contribution and permutation importance of each variable, permutation importance measures the drop in model performance when a variable’s values are randomly shuffled, reflecting its predictive contribution. We gain a clearer picture in Figure 3 of which climatic factors are most influential in determining suitable habitats for *P. aubei*.

The percent contribution and permutation importance of various bioclimatic variables in the MaxEnt model for predicting habitat suitability of *P. aubei* can be seen in Table 1. The bio12 (annual precipitation) and bio6 (minimum temperature of the coldest month) are the most influential variables, with contributions of 30.4% and 29%, respectively. Their permutation importance values (27.3% and 22.8%) confirm their key role. Other factors like bio15 (precipitation seasonality) and bio8 (mean temperature of the wettest quarter) also contribute, though less significantly. Variables such as bio1 (annual mean temperature), bio13 (precipitation of the wettest month), and bio3 (isothermality) have minimal influence.

To further assess the importance of individual environmental variables in predicting the habitat suitability of *P. aubei*, the Jackknife test was conducted using the MaxEnt model. This test evaluates the contribution of each variable by measuring the change in model performance when a specific variable is omitted or used alone. The following Figure 4 presents the results of this test, providing a visual representation of how each environmental factor influences the model’s predictive ability.

The Jackknife test of variable importance for the MaxEnt model of *P. aubei* [22] illustrated the contribution of different environmental variables to the species’ habitat suitability in Figure 3. From the results, it is evident that bio6 (minimum temperature of coldest month) and bio1 (annual mean temperature) play significant roles in determining habitat suitability, as their omission leads to a noticeable decrease in training gain. Additionally, bio8 (mean temperature of the wettest quarter) and bio13 (precipitation of the wettest month) contribute substantially. The blue bars for these variables indicate that they hold predictive power even when used individually.

### 3.2. The Effect of Temperature on Developmental Duration

In contrast, altitude (alt), bio15 (precipitation seasonality), and bio3 (isothermality) show relatively lower individual contributions, as seen from their short blue bars. However, their absence still impacts the model, suggesting they provide complementary information. The red bar, representing the model trained with all variables, achieves the highest training gain, indicating that incorporating multiple environmental factors enhances prediction accuracy.

Overall, the Jackknife test confirms that temperature and precipitation-related variables are the most influential factors in predicting the potential distribution of *P. aubei*, emphasizing the importance of climatic conditions in its habitat suitability analysis.

To gain a deeper understanding of how environmental factors influence the habitat suitability of *P. aubei*, response curves were generated for key variables in the MaxEnt model. These curves demonstrate how the likelihood of species occurrence changes concerning variations in environmental conditions.

The response curves of eight environmental variables used in the MaxEnt model for predicting the habitat suitability of *P. aubei* are shown in Figure 4. These response curves illustrate how the probability of species presence changes as a function of each environmental variable, providing insight into the species’ ecological preferences. Each graph represents a single environmental variable, with the x-axis denoting the variable’s value and the y-axis showing the logistic output of the MaxEnt model, which indicates habitat suitability. The blue-shaded areas reflect variability in the model predictions, while the red lines represent the mean response. From the response curves, it is evident that certain variables have a strong nonlinear influence on suitability possibility. The blue-shaded areas reflect variability in model predictions, while the red lines represent the mean response. From the response curves, it is evident that certain variables have a strong nonlinear influence on habitat suitability. Some graphs show a steep decline or peak at specific values, indicating environmental thresholds where the species is most or least likely to occur. Overall, these response curves help identify the critical environmental conditions affecting *P. aubei*’s distribution, highlighting the species’ sensitivity to changes in temperature, precipitation, and other ecological factors.

### 3.3. Prediction of Potential Geographic Distribution of P. aubei Under Current Climatic Conditions

The MaxEnt model predicted the potential geographic distribution of *P. aubei* under current climatic conditions [23].

Under current conditions, suitable areas cover 331.97 × 10^4^ km^2^ (Figure 5). Future projections show increases across all SSPs, especially SSP5-8.5 in the 2050s, where high-suitability areas will rise by 82.29%. Taiwan and humid subtropical mainland zones are particularly vulnerable. Regionally, unsuitability is projected in western, northern, and southernmost parts of China, such as Tibet, Xinjiang, Qinghai, Inner Mongolia, Heilongjiang, Jilin, Hainan, and Guangdong. Low-suitability areas include Guangxi, Jiangxi, Hunan, Jiangsu, and Fujian. Medium-suitability regions are concentrated in Yunnan, Guizhou, Hubei, and Anhui. High-suitability zones are primarily located in central China, including eastern Sichuan, and cover almost all of Shandong, as well as Beijing and Tianjin.

The high-suitability zones are primarily located in warm, temperate, and subtropical regions, where climatic factors such as temperature, precipitation, and humidity create favorable conditions for the species. Taiwan exhibits climatically analogous zones to mainland China’s southeastern regions despite the geographic separation. The predicted habitat suitability suggests that *P. aubei* is more likely to be established in humid, forested, and cultivated areas, particularly where host trees, such as cypress species, are abundant.

The predicted changes in suitable areas for *P. aubei* under different climate scenarios (SSP1-2.6, SSP2-4.5, and SSP5-8.5) for the 2050s and 2090s can be seen in Table 2. Under current conditions, the total suitable area is 331.97 × 10^4^ km^2^, with 203.78 × 10^4^ km^2^ of low suitability, 91.31 × 10^4^ km^2^ of medium suitability, and 36.88 × 10^4^ km^2^ of high suitability. The table shows that under future climate scenarios, suitable areas are expected to increase, particularly in medium- and high-suitability regions. The largest increase is projected under the SSP5-8.5 scenario for the 2050s, with the area of high suitability rising by 82.29%. Consequently, low-suitability regions generally decrease, especially under the SSP2-4.5 scenario for the 2090s, where a reduction of 4.43% is seen.

### 3.4. Potential Habitat Changes of P. aubei Under Future Climate Scenarios

This study not only predicts the current distribution of suitable habitats for *P. aubei* but also forecasts its habitat suitability for the 2050s and 2090s, as depicted in Figure 6. Future climate change is expected to have a mix of positive and negative impacts on different habitats. Overall, there will be a slight increase in the total of suitable regions, with the range spanning from 0.47% to 6.84% by the 2050s. However, when it gets closer to the 2090s, the relative total suitable regions are expected to be between 2.90% and 3.68%. When looking at specific habitats, there will be a decrease in low-habitat-suitability regions, with a change of −4.73% by the 2050s.

However, there will be an increase in high-habitat-suitability regions, with a change of 82.29% by the 2050s, which can be seen in Table 2. In the 2090s, the change in high-habitat-suitability regions will be 44.18%. The expansion of highly suitable areas under all scenarios, especially SSP5-8.5, suggests the potential for *P. aubei* to thrive in new regions. A decrease in low-suitability areas may lead to habitat loss in some regions. Medium suitability shows variability, requiring careful monitoring. It is important to note that these projections assume static distributions of host trees (e.g., *Cupressus* spp.), which are critical for *P. aubei* survival. Climate change may alter the suitability and geographic range of these host species, potentially reshaping the beetles’ habitat availability. For instance, if host trees experience range contractions or shifts due to changing climatic conditions, *P. aubei*’s projected expansion into new regions could be constrained. Future studies integrating dynamic host tree distribution models under climate scenarios would enhance the accuracy of predictions and provide a more holistic understanding of ecosystem-level impacts.

### 3.5. Centroid Changes in Potential Distribution

Initially, the centroid moves southeastward across all scenarios, suggesting a general trend toward higher latitudes and elevations [24].

However, in later periods, the direction of movement varies depending on the emission scenario.

Under SSP1-2.6, the centroid instead shifts toward the northeast at a slower pace than the initial movement. This shift occurred under high-emission scenarios (SSP5-8.5) due to warming-induced precipitation changes in southwestern China. In contrast, under SSP2-4.5 and SSP5-8.5, the centroid exhibits a southwestward shift, indicating a reversal in habitat suitability trends. The extent of this shift is more pronounced in SSP5-8.5, suggesting a stronger response to higher emission levels (Figure 7).

The observed centroid movements reflect changes in climatic factors affecting habitat suitability, with variations in temperature, precipitation, and other environmental conditions influencing the potential distribution of *P. aubei* over time.

## 4. Discussion

The findings from this study identified annual precipitation (bio12) and the lowest temperature of the coldest month (bio6) as the two most significant environmental factors influencing the distribution of *P. aubei* in China. These variables contributed 30.4% and 29%, respectively, to the predictive model. Such results are consistent with similar studies on the distribution of forest pests, which often emphasize the role of temperature and precipitation in shaping species’ ecological preferences. For instance, it was reported that temperature and precipitation were key determinants in modeling the distribution of *Cunninghamia lanceolata*, a conifer species affected by pests, highlighting a shared ecological relationship between climate factors and pest distributions [25]. Similarly, it was found that temperature and humidity are significant in determining the range of bark beetles like *P. aubei armatus* [22]. The prominence of these factors in our study reinforces the broader ecological pattern observed in which *P. aubei* and related species exhibited strong climate-driven distribution shifts.

Application of the response curves and the Jackknife test reveals that *P. aubei* is highly sensitive to both temperature and precipitation, displaying narrow ecological tolerances. (Figure 4). Habitat suitability for the species decreases significantly when these variables exceed certain thresholds, supporting previous research highlighting the vulnerability of bark beetles to climate extremes. *P. aubei*’s suitability declines sharply when bio12 exceeds 1500 mm or bio6 drops below 5 °C (Figure 4). The southwestward shift under high emissions highlights complex climate interactions. Similarly, it was found that many bark beetle species have restricted ecological thresholds, indicating their sensitivity to changes in climatic conditions. For example, we discussed the narrow ecological tolerances of bark beetles, which are closely tied to temperature and moisture levels. These findings suggest that climate change could drastically affect the range and abundance of *P. aubei* and other similar pests, necessitating a focused effort in climate change monitoring for pest management [26].

The projections generated under various Shared Socioeconomic Pathways (SSPs) indicate an overall increase in the habitat suitability for *P. aubei*, especially under the SSP5-8.5 scenario, with a predicted rise of 82.29% in suitable areas by the 2050s. This suggests that under future climate scenarios, the species may expand into regions that were previously unsuitable for its survival. These results align with previous studies that have predicted the northward expansion of forest pests due to climate warming; for example, there are similar trends for other pest species, with a shift in habitat suitability toward higher latitudes under warming scenarios [27]. The northeastward shift observed in this study further corroborates the findings from, we highlighted the general tendency for forest pests to migrate poleward in response to climate change. Limitations include neglecting dispersal barriers and host dynamics [25].

However, a unique finding of this study was the southwestward shift observed under specific scenarios (SSP2-4.5 and SSP5-8.5) in later periods. This reversal of expected trends adds complexity to the understanding of future pest distributions, as higher levels of emissions could potentially lead to unpredictable changes in habitat suitability. Such findings underscore the necessity for adaptive management approaches that can account for varying levels of emissions and their unpredictable impacts on pest distribution.

Figure 7’s centroid shift analysis conducted in this study reveals a northeastward shift in the habitat suitability of *P. aubei* under all climate scenarios, reflecting a common migration pattern of species toward higher latitudes and altitudes as temperatures rise [28]. This trend aligns with global patterns of species migration in response to climate change. However, the southwestward shift observed under the SSP2-4.5 and SSP5-8.5 scenarios in the later decades indicates that greater emission levels may cause more complex and less predictable ecological responses. This result contrasts with more straightforward migration predictions seen in other studies (Table 3), such as those that noted the poleward shift of many species in response to warming [29]. The divergence observed in this study highlights the complexities of species migration and the potential for unexpected shifts in habitat suitability, emphasizing the need for adaptive management strategies that are flexible and can respond to rapidly changing climatic conditions.

### Challenges and Future Work

This study faced several challenges in predicting the spread of *P. aubei*. The MaxEnt model assumes unlimited dispersal, overlooking real-world barriers to movement. It also does not account for interactions with host species or land-use changes, limiting its accuracy [46]. Additionally, reliance on sparse occurrence data may not capture the full range of the beetles, especially in under-studied regions. Future climate projections are uncertain due to variable climate models and emission scenarios, making the actual spread of *P. aubei* potentially different from predictions. These challenges highlight the need for more refined models and better data integration to improve management strategies. Future models can be improved by integrating dispersal barriers, such as physical obstacles and ecological factors, to better reflect the species’ movement limitations and include host tree distribution and interactions to more accurately predict habitat suitability for *P. aubei*. Land-use data should be integrated to assess how urbanization, agriculture, and deforestation may impact the beetles’ spread and habitat availability. More detailed and widespread occurrence data should be collected to enhance the accuracy of predictions, especially in regions where *P. aubei* has not yet been reported. Adaptive management strategies should be developed based on dynamic modeling to address the unpredictable spread of *P. aubei* under climate change. The centroid analysis of *P. aubei*’s potential distribution indicates a shift in habitat suitability under different climate scenarios [47]. Future work can incorporate AI integration and multi-modal approaches to enable more robust and accurate predictions [48].

## 5. Conclusions

In conclusion, this study successfully utilized the MaxEnt model to predict the current and future distribution of *P. aubei* under varying climatic scenarios. The findings underscore the significant role of key bioclimatic variables, particularly temperature and precipitation, in determining habitat suitability for the species. With climate change, the species is expected to expand into new regions, particularly under high-emission scenarios like SSP5-8.5, where high-suitability areas may increase dramatically. These shifts in habitat suitability pose potential risks to forest ecosystems, emphasizing the importance of proactive forest management strategies.

This study also highlights the need for early detection, continuous monitoring, and adaptive management strategies to mitigate the ecological impacts of *P. aubei*’s spread. By forecasting potential habitat changes under different climate scenarios, the research provides valuable insights that can inform conservation efforts and guide policy decisions to protect vulnerable ecosystems from invasive pests. Although the model provides strong predictions, further research incorporating other factors, such as host tree distribution and land-use changes, would enhance the robustness of the predictions and help improve pest management strategies. Ultimately, this research lays the groundwork for more effective monitoring and adaptive management in the face of climate change. Proactive monitoring in high-risk regions (SSP5-8.5 zones) is critical. Integrating host tree data will improve future models.

## Figures and Tables

**Figure 1 insects-16-00576-f001:**
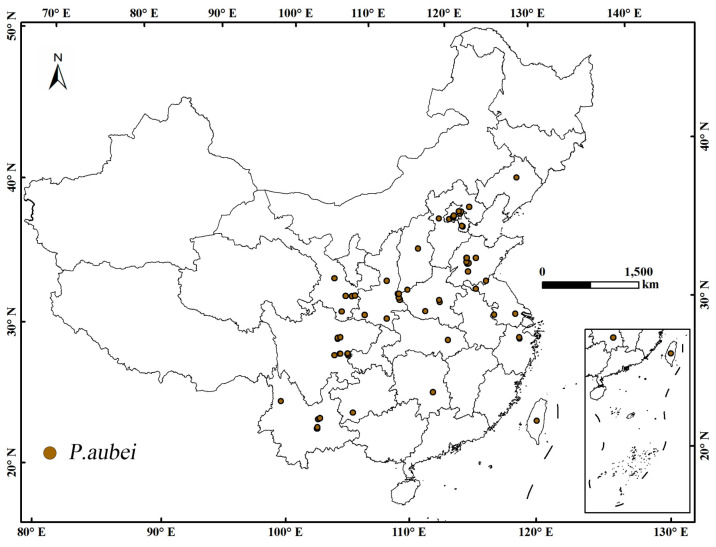
Occurrence records of *Phloeosinus aubei* in China based on GBIF data. The main map shows distribution points (brown dots) across latitudes 20° N–50° N and longitudes 80° E–140° E. An inset highlights island regions.

**Figure 2 insects-16-00576-f002:**
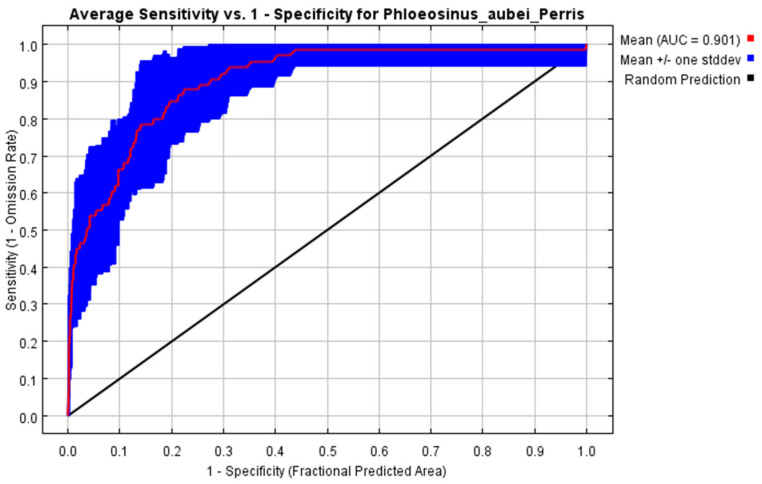
Receiver Operating Characteristic (ROC) curve for *Phloeosinus aubei*. Habitat suitability model.

**Figure 3 insects-16-00576-f003:**
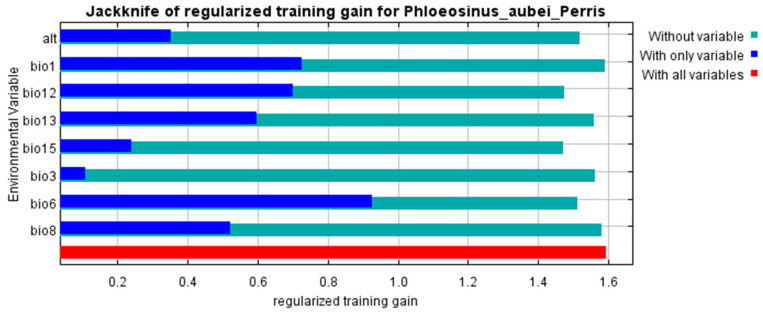
Jackknife test of variable importance for the MaxEnt model of *Phloeosinus aubei*.

**Figure 4 insects-16-00576-f004:**
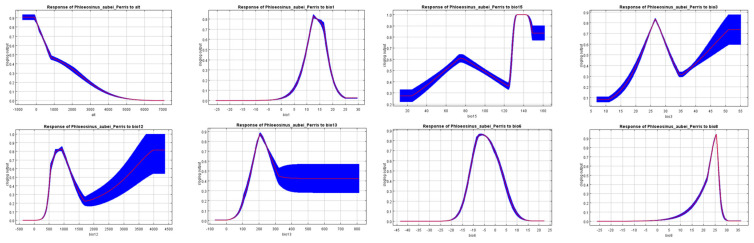
Response curves of eight environmental variables.

**Figure 5 insects-16-00576-f005:**
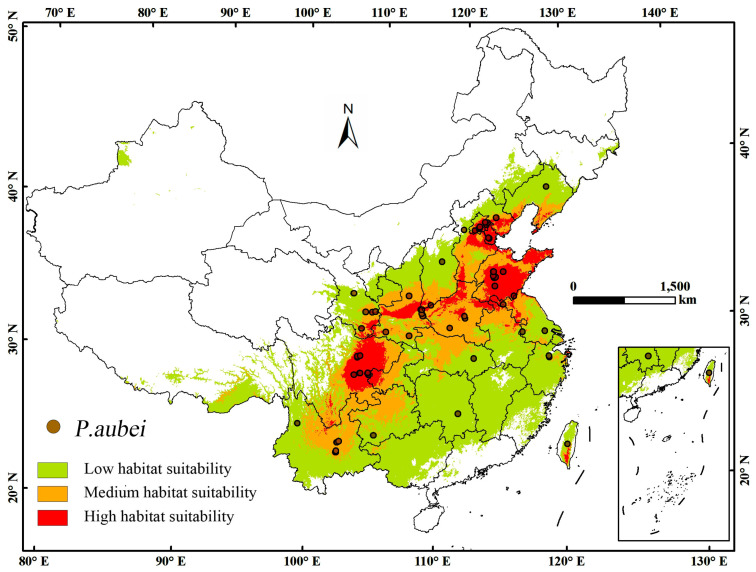
Distribution of suitable habitats for *Phloeosinus aubei* under the current climate.

**Figure 6 insects-16-00576-f006:**
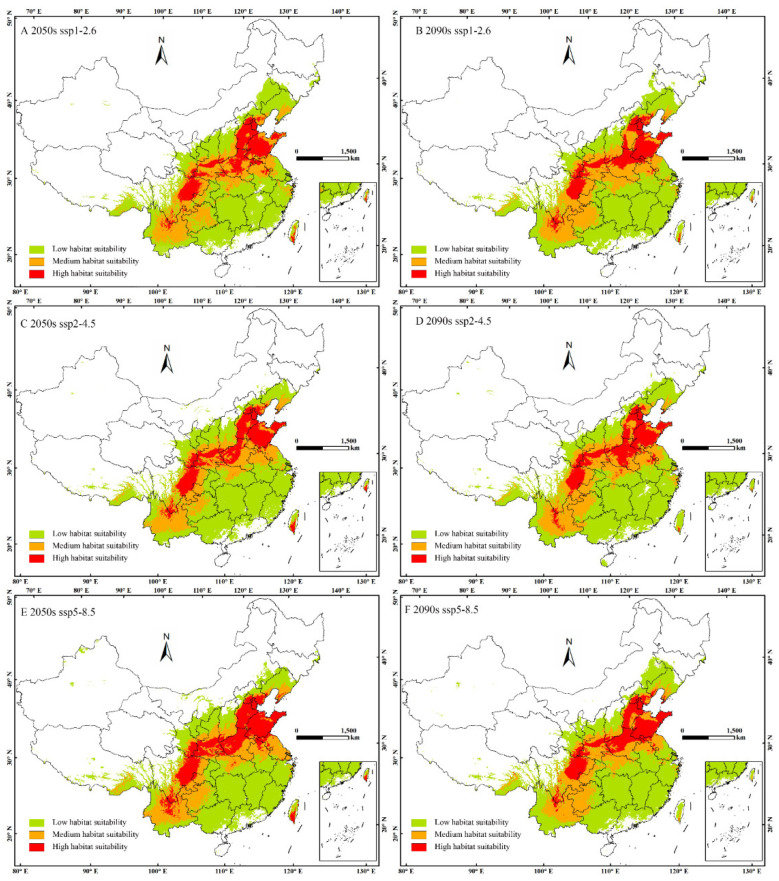
Habitat suitability for *Phloeosinus aubei* under future climate scenarios.

**Figure 7 insects-16-00576-f007:**
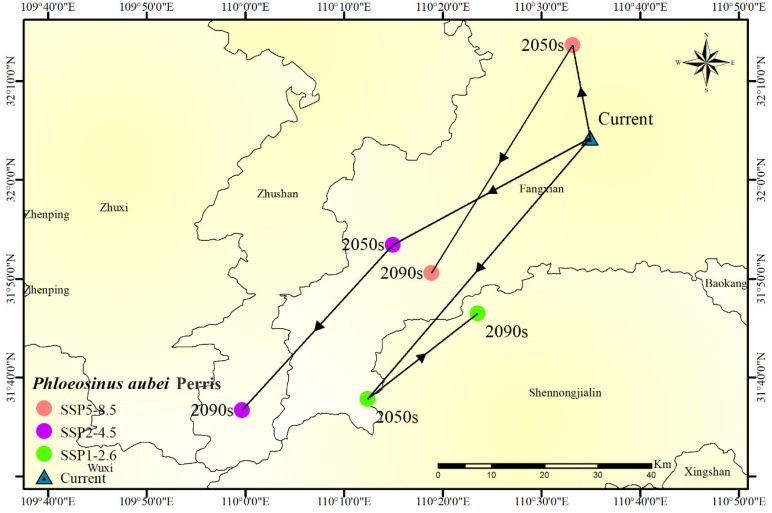
Changes in the centroid of the potential highly suitable distribution of *Phloeosinus aubei* in a southwestward direction.

**Table 1 insects-16-00576-t001:** Contribution and permutation in estimating climate variables in the MaxEnt model of *Phloeosinus aubei*.

Abbreviation	Description	Percent Contribution	Permutation Importance
bio12	Annual Precipitation (mm)	30.4%	27.3%
bio6	Minimum Temperature of the Coldest Month (°C)	29.0%	22.8%
bio15	Precipitation Seasonality (Coefficient of Variation)	7.1%	8.2%
bio8	Mean Temperature of Wettest Quarter (°C)	4.5%	8.2%
bio3	Isothermality (B102/B107) × 100	2.0%	1.1%
bio13	Precipitation of Wettest Month (mm)	0.5%	12.7%
bio1	Annual Mean Temperature (°C)	0.4%	2.2%
alt	Altitude/Elevation	0.1%	2.8%

**Table 2 insects-16-00576-t002:** Prediction of suitable areas for *P. aubei* under current and future climatic conditions.

Scenarios	Decade	Total Suitable Regions		Regions of Low Habitat Suitability		Regions of Medium Habitat Suitability		Regions of High Habitat Suitability	
		Area (10^4^ km^2^)	Area Change (%)	Area (10^4^ km^2^)	Area Change (%)	Area (10^4^ km^2^)	Area Change (%)	Area (10^4^ km^2^)	Area Change (%)
-	current	331.97	-	203.78	-	91.31	-	36.88	-
SSP1-2.6	2050s	342.04	3.03%	201.16	−1.28%	93.02	1.88%	47.86	29.76%
	2090s	341.60	2.90%	194.24	−4.68%	100.42	9.98%	46.94	27.28%
SSP2-4.5	2050s	333.54	0.47%	194.75	−4.43%	87.08	−4.64%	51.72	40.23%
	2090s	344.14	3.67%	196.41	−3.62%	97.69	6.98%	50.05	35.71%
SSP5-8.5	2050s	354.67	6.84%	194.14	−4.73%	93.30	2.18%	67.23	82.29%
	2090s	344.20	3.68%	199.52	−2.09%	91.50	0.21%	53.17	44.18%

**Table 3 insects-16-00576-t003:** Comparison of key aspects between this research and similar research.

Aspect	*P. aubei*	Similar Research	Key Differences
Key Environmental Variables	bio12 (precipitation: 30.4%) and bio6 (temperature: 29%) were most significant.	[30,31,32,33] Temperature and precipitation are consistently identified as critical for pest distribution.	Quantifies exact contributions (30.4% and 29%), while others discuss general trends without specific percentages.
Modeling Approach	MaxEnt was used with response curves and Jackknife tests to assess narrow tolerances.	[34,35,36,37] MaxEnt is widely applied, but (9) critiques its assumptions (e.g., unlimited dispersal).	Explicitly addresses *P. aubei*’s narrow tolerances, whereas (9) focuses on methodological limitations.
Future Projections (SSPs)	SSP5-8.5 predicts an 82.29% habitat suitability increase by the 2050s; a unique southwestward shift under high emissions.	[38,39,40] Northward shifts are common, but the IPCC notes regional variability.	Identifies a counterintuitive southwestward shift not highlighted in other pest studies.
Ecological Implications	*P. aubei*, as a secondary invader, exacerbates tree mortality in stressed forests.	[41,42,43] Bark beetles are generally linked to forest stress and economic losses.	Emphasizes *P. aubei*’s specific role in China while reviewing European outbreaks.
Management Strategies	Recommends early detection, public awareness, and adaptive management tied to SSPs.	[44,45] Supports rapid response and citizen science.	Integrates climate-specific adaptation (e.g., emission scenario-based strategies), unlike general focus.
Limitations	Notes MaxEnt’s neglect of dispersal barriers/host interactions and suggests incorporating land-use data.	[44,45] Critiques MaxEnt’s simplicity but proposes broader variable integration.	Links limitations to *P. aubei*’s dispersal ecology and provides generic recommendations.

## Data Availability

The original contributions presented in this study are included in the article. Further inquiries can be directed to the corresponding author.

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
