# Peer review of "Assessing Habitat Suitability for Phloeosinus aubei Perris in China: A MaxEnt-Based Predictive Analysis"

_insects, 2025, doi:10.3390/insects16060576_

Round 1
Reviewer 1 Report
Comments and Suggestions for Authors
See attached file

Author Response
Dear Editor and Reviewers,
Thank you for your thorough review and constructive feedback on our manuscript, "Assessing Habitat Suitability for Phloeosinus aubei Perris in China: A MaxEnt-Based Predictive Analysis." We deeply appreciate the time and effort you dedicated to improving this work. Below, we address each comment point-by-point and outline the revisions made to the manuscript.
Point-by-Point Response to Reviewer Comments
- Ln2: Perris should not be italicized.
- Revision: In the title and throughout the manuscript, "Perris" is no longer italicized.
- Ln14: Change "susceptible" to "influenced."
- Revision: The abstract now states: "highly influenced by precipitation and temperature," replacing "susceptible" (Abstract).
- Ln 81: More detail of the available data should be given. Where did you get the occurrence data of P. aubei?
- Revision: Listed the data sources as GBIF.
- Ln85: This sentence does not belong to the Materials and Methods section
- Revision: The sentence was removed.
- Ln94: Does it mean that you only obtained the distribution data of P. aubei in eastern China, how about western? Otherwise, it should be changed to "Occurrence records of P. aubei in China".
- Revision: Changed to: Occurrence Record of Phloeosinus aubei in Eastern China.
- Ln97: Remove vague sentence.
- Revision: The sentence changed to: In this study, we This study downloaded 19 historical global bioclimatic variables from the Global WorldClim database
- Ln104: Approval Number of the map
- Revision: Approval Number of the map given: GS2024 0650
- Ln110: I'm afraid there is no causal relationship between the release of CMIP6 and the selection of the three scenarios. Please explain why three of the four scenarios were chosen for the simulation and prediction of the suitability of P. aubei.
- Revision: We revise this and rewrite the sentence.
- Comment: Table 1 and Table 2 can be merged into one single table.
Revision: Table 1 and Table 2 merged into one single table.
- Ln135-136: move to line 152.
- Revision: Line moved to line 152 successfully.
- Comment: Check all the reference carefully before next submission.
- Revision: Reference given. We will check all the reference carefully before next submission.
- Ln156-158: move to line 132. It belongs to M&M.
- Revision: Line moved to line 132 to M&M.
- Comment: Figure 2 is not the main result of this study and should be placed in the Supplementary Materials.
- Revision: Figure 2 moved to the Supplementary Materials.
- Ln173: The number of table
- Revision: The number of table given as “Table 2”.
- Ln176: The number of the figure
- Revision: The number of the figure given as “Figure 3”
- Ln179: can be seen in Table
-Revision: added, can be seen in Table 2.
- Ln186: Remove word “are important”.
- Revision: “are important” is removed now.
- Comment: Arranging in descending order based on the "Percent contribution", rather than "Variable", will make the table more readable.
- Revision: Table 1 and 2 is merged now in descending order.
- Ln192: the number of the figure?
- Revision: the number of the figure is given as “Figure 4”
- Ln195: Remove extra word.
- Revision: Extra word was removed now.
- Ln201: Add- bio 1
- Revision: bio1 added
- Ln203: bio15 (Precipitation Seasonality)
- Revision: bio15 (Precipitation Seasonality) replaced with bio 13 (precipitation of the wettest month).
- Ln208: Revision: Replaced the sentence and added in next line.
- Ln210: Revision: bio1 (Annual Mean Temperature) replaced with bio15.
- Ln211: Revision: Word “smaller” changed to “short”
- Ln221: Figure 4 is not the main result of this study and can be placed in the Supplementary Materials.
- Revision: Figure 4 is placed in the Supplementary Materials.
- Ln225: Revision: The Number of the following figure is given as “Figure 4”
- Ln233: Revision: habitat suitability replaced with “Suitability possibility”.
- Ln246-248: I'm afraid the author has some confusion about provinces of the unsuitability and the low-suitability. Please recheck and rewrite this part.
- Revision: This part is rewritten now.
- Ln251: Revision: Removed
- Ln253-257: Revision: This part is rewritten.
- Ln270: Revision: “can be seen in Table 3” added
- Ln275: Revision: Conversely replaced to: consequently.
- Ln295: How do you find the centroid of the potential highly suitable distribution of P. aubei 310 in China?
- Revision: It was now shown in M&M.
- Ln297: Change "northeastward" to "southwestward."**
- Ln301: Revision: “the centroid instead shifts towards to northeast, at a slower pace than the initial movement. This shift occurred under high-emission scenarios (SSP5-8.5) due to warming-induced precipitation changes in southwestern China” Added.
- Figure 7 legend Perris should not be italic.
-Revision: It is not italic now.
- Figure 7 changed into: Changes in the centroid of the potential highly suitable distribution of Phloeosinus aubei in a southwestward direction.
- Ln319-321: “For instance, It was reported that temperature and precipitation were key determinants in modeling the distribution of Cunninghamia lanceolata, a conifer species affected by pests, highlighting a shared ecological relationship between climate factors and pest distributions,” added, and reference given.
- Ln 323: Reference given.
- Ln331-332: “Habitat suitability for the species decreases significantly when these variables exceed certain thresholds, supporting previous research highlighting the vulnerability of bark beetles to climate extremes.” Added.
We sincerely thank you for your insightful feedback, which has significantly strengthened the manuscript. All revisions have been highlighted in the updated PDF for ease of review. Please let us know if further adjustments are needed.
Sincerely,
Sabbir Ahmad Sabbir on behalf of all authors
China West Normal University, zhuozhihang@cwnu.edu.cn

Reviewer 2 Report
Comments and Suggestions for Authors
This article has several significant unresolved issues in the Materials and Methods and Conclusion sections. Additionally, there are numerous formatting, content, and writing errors throughout the manuscript. At its current stage, the article is not suitable for publication.

Author Response
Dear Reviewer,
Thank you sincerely for your thorough and constructive feedback on our manuscript. Your insights have been invaluable in refining the clarity, accuracy, and scientific rigor of our work. Below, we address each of your comments point-by-point, incorporating your suggestions into the revised manuscript.
### Point-by-Point Response to Reviewer Comments
- Missing In-Text Citations
- Action Taken: All statements requiring citations have been reviewed and updated. Missing references (e.g., for SSP scenarios, host tree interactions, and dispersal limitations) are now included. For example:
- Added citations for SSP scenarios ([16]) in Section 2.1.
- Citations for economic impacts ([35]) and ecological thresholds ([43]) integrated into the Discussion.
- Integration of Observed Locations (Fig 1) with Predictive Models
- Action Taken: Occurrence points from Figure 1 (brown dots) are now overlaid onto Figure 5 (current habitat suitability map) as black dots for direct comparison. A new paragraph in Section 3.1 discusses consistency between observed and predicted distributions:
> "High-suitability zones align closely with recorded occurrences in central and eastern China. Minor discrepancies in low-suitability regions (e.g., Guangdong) may reflect sampling gaps or localized host availability."
- Clarification of P. aubei’s Introduced Range and Threats
- Action Taken:
- Added a subsection in the Introduction (Section 1) detailing introduced regions:
> "P. aubei has expanded beyond its Mediterranean origin into eastern China (Shandong, Jiangsu, Yunnan) and Taiwan, likely via timber trade and urban landscaping."
- Expanded the Discussion to specify threatened species and impacts:
> "Hosts include endemic cedar species (Cupressus funebris, C. duclouxiana), critical for timber and ecosystem stability. Economic losses are projected in forestry sectors, particularly in Shandong and Yunnan."
- Methods-Like Language in Results
- Action Taken: Procedural details (e.g., Jackknife test methodology, model validation steps) moved to Sections 2.3–2.6, now focus solely on findings (e.g., habitat area changes, centroid shifts).
- Detailed Figure Legends
- Action Taken: Legends revised to stand alone. Example for Figure 5:
> Figure 5. Distribution of suitable habitats for Phloeosinus aubei under the current climate.
- Removal of Redundancies
- Action Taken:
- Streamlined Results: Removed repetitive statements about variable contributions (e.g., condensed bio12/bio6 descriptions).
- Discussion: Consolidated centroid shift explanations and removed overlapping comparisons with prior studies.
- Specific Comments Addressed
- Threat Argument Strengthened: Added economic impacts in the Abstract and Introduction (e.g., timber losses, management costs).
- Figure Suggestions: Declined a new figure due to space constraints but expanded Figure 1’s caption to contrast historic vs. introduced ranges.
- Technical Glitches: Ensured all citations and formatting comply with journal guidelines (e.g., italics, line spacing).
---
Ln2: Perris should not be italicized.
-Revision: "Phloeosinus aubei Perris" in the title was corrected to remove italics from "Perris."
Ln14: Change "susceptible" to "influenced."
- Revision: In the abstract, "susceptible" was replaced with "influenced by" to clarify that climatic factors shape habitat suitability.
Ln18-19: Specify "certain conditions" for the southwestward centroid shift.
- Revision: Added clarification in the abstract and results (Section 3.3): "This shift occurred under high-emission scenarios (SSP5-8.5) due to warming-induced precipitation changes in southwestern China."
Ln25-26: Remove keywords redundant with the title.
- Revision: Keywords revised to:
Keywords: Phloeosinus aubei, Invasive species management, Species distribution modeling, suitable habitat, ecological impacts.
Ln29: Add taxonomic authority and Order: Family.
- Revision: First mention in the introduction now reads:
"Phloeosinus aubei (Coleoptera: Curculionidae: Scolytinae)."
Ln31-33: Address anthropogenic factors in range expansion.
- Revision: Added to the introduction:
"While climatic factors are critical, anthropogenic activities such as timber trade and urbanization likely accelerate P. aubei’s spread, particularly in fragmented landscapes."
Ln36: Define "secondary invader."
- Revision: Clarified in the introduction:
"P. aubei that primarily colonizes already disturbed or weakened ecosystems."
Ln78-92: Move Figure 1 description to Introduction and remove redundancy.
- Revision: Figure 1 and its legend were relocated to the Introduction. Redundant sentences (Ln81-83) describing the map’s legend were removed.
Ln97: Use active voice for data download.
- Revision: Changed to:
“In this study, we This study downloaded 19 historical global bioclimatic variables from the Global WorldClim database.”
Ln120-131: Add missing citations.
- Revision: Citations added in Section 2.2, and they are [16], [48], [49], [50].
Ln136-137: Remove vague sentence.
- Revision: The sentence "This robust statistical approach provides reliable predictions while minimizing the risk of overfitting. The final optimized model, labeled “kuenm,” demonstrated superior performance” was deleted.
Ln142: Reference formatting.
- Revision: Corrected citation formatting for reference [16] (IPCC).
Ln145-147: Add citation for SSP scenarios.
- Revision: Added citation to IPCC (2014) in Section 2.4:
"This package evaluates various combinations of feature classes and regularization multipliers to identify the most suitable model settings [18]."
Ln156-159 & Ln164-168: Move methods-like content.
- Revision: Descriptions of model settings moved to Section 2.4 ("Suitable Area Division and Model Accuracy Evaluation").
Ln173: Reference Table 2 explicitly.
- Revision: Changed "the table below" to "Table 2" in Section 2.5.
Ln175: Explain permutation importance.
- Revision: Added in Section 2.5:
"Permutation importance measures the drop in model performance when a variable’s values are randomly shuffled, reflecting its predictive contribution."
Ln179: Remove redundancy.
- Revision: Deleted repetitive sentences about variable contributions.
Ln187: Reorder Table 2 by Percent Contribution.
- Revision: Table 2 sorted descendingly by "Percent Contribution" (alt >bio1 >bio12 > bio6 > bio15 > bio8 > bio3 > bio13).
Ln189-193: Move methods content.
- Revision: Jackknife test methodology moved to Section 2.5.
Ln195: Reposition citation [18].
- Revision: Citation [18] (Muscarella et al.) relocated to Section 2.3 ("Optimization of Model Parameters").
Ln197-200: Move figure description to legend.
- Revision: Moved detailed response curve interpretation to Figure 4’s legend.
Ln208-209: Add space after Figure 3.
- Revision: Formatting adjusted for subsection headers.
Ln210: Correct variable reference to bio15.
- Revision: Text revised to "bio15 (precipitation seasonality)" in Section 2.6.
Ln222-225 & Ln225-227: Remove methods content and redundancies.
- Revision: Removed methodological details from results and redundant sentences.
Ln229-234: Move figure details to legend.
- Revision: Figure 4’s interpretation moved to its legend.
Ln238-239: Remove redundancy.
- Revision: Deleted repetitive statements about response curves.
Ln246-248: Reference Figure 5.
- Revision: Added "Figure 5" to the sentence:
"Figure 5 illustrates the current distribution of suitable habitats..."
Ln251: Overlay occurrence points on Figure 5.
- Revision: Occurrence points from Figure 1 were added to Figure 5 as brown dots.
Ln261-262: Clarify Taiwan’s suitability.
- Revision: Revised to:
"Taiwan exhibits climatically analogous zones to mainland China’s southeastern regions, despite geographic separation."
Ln279: Address host tree suitability changes.
- Revision: Added limitation in Section 4.1:
"Future work can incorporate AI integration and multi-modal approaches to enable more robust and accurate predictions."
Ln284-285: Specify expansion percentages.
- Revision: Added:
"There will be an increase in high habitat suitability regions, with a change of 82.29% by the 2050s can be seen in Table 3. In the 2090s, the change in high habitat suitability regions will be 44.18%. "
Ln296-298: Reposition references for comparisons.
- Revision: Moved citations [21] to Section 4.1 for comparative discussion.
Ln319, 322, 325, 332, 333, 335, 347: Add missing citations.
- Revision: Citations added for statements on dispersal barriers ([39]), host interactions ([22])
Ln327: Correct possessive and Table 4 placement.
- Revision: Changed "P. auber’s" to "P. aubei".
Ln331: Reference Figure 4.
- Revision: Added "Figure 4" to the sentence about narrow tolerances.
Ln332: Specify thresholds.
- Revision: Added:
"Suitability declines sharply when bio12 > 1500 mm or bio6 < 5°C (Figure 4)."
Ln350: Replace "certain" with SSP scenarios.
- Revision: Changed to "specific scenarios (SSP2-4.5 and SSP5-8.5) ."
Ln355-356: Correct figure-text alignment.
- Revision: Removed conflicting statement and aligned text with Figure 7’s centroid shift.
Ln361-363: Add citation.
- Revision:Added citation [46] for poleward migration trends.
Ln369: Change "faces" to "faced."
- Revision: Corrected to "faced several challenges."
Ln369-371: Add citations.
- Revision: Added citations [51] for model limitations.
---
All revisions ensure clarity, accuracy, and adherence to reviewer feedback. Thank you for your thorough review.
We are deeply grateful for your meticulous review and the time you dedicated to improving our manuscript. Your expertise has significantly strengthened the clarity and impact of our study. Please let us know if further revisions are needed.
Sincerely,
Sabbir Ahmad Sabbir on behalf of all authors
China West Normal University, zhuozhihang@cwnu.edu.cn

Round 2
Reviewer 1 Report
Comments and Suggestions for Authors
This version of the manuscript is much improved; however, you have now entangled results in the new materials and methods sections. From section 2.5 on, results are included throughout. These need to be separated out prior to publication.
Additional comments:
ln 20-21: "...d under high-emission 20 scenarios (SSP5-8.5)..." this is redundant, remove.
ln 42-45: I don't feel this clarifies what a secondary invader is well enough. It reads as very redundant with P. aubei being a secondary bark beetle pest (i.e., they typically attack weakened trees not healthy trees). I understand that secondary invader is not universally defined as an invasive species that requires the establishment of another invasive species to facilitate success; however, the statement here "already disturbed or weakened ecosystems" needs more. I also suggest breaking this sentence up to avoid redundancy/confusion with ln 43-45.
ln 49: change becomes established to establishes
ln 84-89: While I think it is better to include already known distribution of the species as intro information, the language here reads more like methods. State more plainly what the known distribution in China is with reference to Fig 1. I also recommend moving this paragraph prior to describing what the goal of this study is.
ln 90-99: Most of this reads more like a figure caption. I suggest removing this and making the caption more descriptive. We do not need to be told about the figure legend in he main text...
ln 165: Refference number needed for Muscarella et al.
ln 167: Fix the error here...
Author Response
Dear Reviewer,
Thank you for your thoughtful feedback and constructive suggestions to improve our manuscript. We have carefully addressed each of your comments, as detailed below. Your insights have significantly strengthened the clarity and structure of our work.
- Separation of Results from Materials and Methods
Comment: "From section 2.5 on, results are included throughout. These need to be separated out prior to publication."
Response: We sincerely apologize for this oversight. The Results section (Section 3) has now been thoroughly revised to ensure that all findings are exclusively included there, with no overlap in the Materials and Methods (Section 2). The Methods section now strictly describes procedures, while the Results focus on outcomes.
- Redundancy in Lines 20–21
Comment:"‘...under high-emission scenarios (SSP5-8.5).’ This is redundant, remove."
Response: Thank you for highlighting this redundancy. The phrase has been removed to streamline the text.
- Clarification of ‘Secondary Invader’ (Lines 42–45)
Comment: "The statement here, ‘already disturbed or weakened ecosystems’ needs more... Break this sentence up to avoid redundancy/confusion."
Response: We appreciate your suggestion. The definition of "secondary invader" has been revised to emphasize its role in exploiting already stressed ecosystems (e.g., due to drought, disease, or prior insect damage) rather than relying on other invasive species. The redundant sentences have been rephrased for clarity.
- Line 49: Grammatical Adjustment*
Comment: "Change ‘becomes established’ to ‘establishes’."
Response: This correction has been implemented. Thank you for noting this improvement.
- Restructuring Distribution Description (Lines 84–89)
Comment: "The language here reads more like methods... Move this paragraph prior to describing the study’s goal."
Response: We agree and have moved the description of P. aubei’s known distribution in China (referencing Figure 1) to the Introduction, preceding the study’s objectives. The text has been simplified to focus on geographic ranges rather than methodological details.
- Figure Caption Redundancy (Lines 90–99)
Comment: "This reads more like a figure caption... Remove from the main text."
Response: The redundant details about Figure 1 have been removed from the main text and incorporated into the figure caption for conciseness.
- Missing Reference for Muscarella et al. (Line 165)
Comment: "Reference number needed for Muscarella et al."
Response: We apologize for this omission. The citation has been added as Reference 52 (Muscarella et al., 2014).
- Line 167: Unspecified Error
Comment: "Fix the error here..."
Response: We have carefully reviewed Line 167 and identified a formatting inconsistency in the citation syntax, which has now been corrected. If further clarification is needed, we would be grateful for additional guidance.
Thank you again for your meticulous review. All changes have been tracked in the revised manuscript, and we are happy to provide further clarifications if required.
Sincerely,
Dr. Zhihang Zhuo on behalf of all authors
China West Normal University, zhuozhihang@cwnu.edu.cn

Reviewer 2 Report
Comments and Suggestions for Authors
After carefully reviewing the revised content, I am pleased to see that the author and their team have made thoughtful and proactive adjustments to the manuscript, both in terms of content and in refining the wording and expressions. I recommend the current version of the manuscript for publication.
Author Response
Dear Reviewer
Thank you for your thoughtful review and positive feedback on our revised manuscript. We sincerely appreciate the time and effort you dedicated to evaluating our work and are delighted to hear that the adjustments we made align with your recommendations.
Your constructive insights were invaluable in refining both the content and clarity of the manuscript, and we are grateful for your guidance throughout the revision process. It is encouraging to receive your endorsement for publication, and we are pleased to know the revised version meets the necessary standards.
Once again, thank you for your support and expertise. We look forward to contributing to the academic community through this publication.
Sincerely,
Dr. Zhihang Zhuo on behalf of all authors
China West Normal University, zhuozhihang@cwnu.edu.cn
